# Identification of structures for ion channel kinetic models

**Kathryn E. Mangold**[1], **Wei Wang**[2], **Eric K. Johnson**[2], **Druv Bhagavan**[1], **Jonathan D. Moreno**[1,2], **Jeanne M. Nerbonne**[2,3], **Jonathan R. Silva**[1] *

**1** Department of Biomedical Engineering, Washington University in St. Louis, St. Louis, Missouri, United States of America, **2** Department of Medicine, Cardiovascular Division, Washington University School of Medicine, St. Louis Missouri, United States of America, **3** Department of Developmental Biology, Washington University School of Medicine, St. Louis, Missouri, United States of America

* jonsilva@wustl.edu

**Data Availability Statement:** Code and data for all models is available on Github: https://github.com/ silvalab/AdvIonChannelMMOptimizer.

## Abstract

Markov models of ion channel dynamics have evolved as experimental advances have improved our understanding of channel function. Past studies have examined limited sets of various topologies for Markov models of channel dynamics. We present a systematic method for identification of all possible Markov model topologies using experimental data for two types of native voltage-gated ion channel currents: mouse atrial sodium currents and human left ventricular fast transient outward potassium currents. Successful models identified with this approach have certain characteristics in common, suggesting that aspects of the model topology are determined by the experimental data. Incorporating these channel models into cell and tissue simulations to assess model performance within protocols that were not used for training provided validation and further narrowing of the number of acceptable models. The success of this approach suggests a channel model creation pipeline may be feasible where the structure of the model is not specified *a priori*.

## Author summary

Markov models of ion channel dynamics have evolved as experimental advances have improved our understanding of channel function. Past studies have examined limited sets of various structures for Markov models of channel dynamics. Here, we present a computational routine designed to thoroughly search for Markov model topologies for simulating whole-cell currents. We tested this method on two distinct types of voltage-gated cardiac ion channels and found the number of states and connectivity required to recapitulate experimentally observed kinetics. Successful models identified with this approach have certain characteristics in common, suggesting that model structures are determined by the experimental data. Incorporation of these models into higher scale action potential and cable (an approximation of one-dimensional action potential propagation) simulations, identified key channel phenomena that were required for proper function. These methods provide a route to create functional channel models that can be used for action potential simulation without pre-defining their structure ahead of time.

**Funding:** This work was supported by National Institutes of Health NHLBI grants R01HL136553 (JRS), R01HL142520 (JMN), R01HL150637 (JRS and JMN) and T32-HL134635 (KM). This work was also supported by an Amazon Web Services Grant (JDM, JRS). The funders had no role in study design, data collection and analysis, decision to publish, or preparation of the manuscript.

**Competing interests:** The authors have declared that no competing interests exist.

## Introduction

Discrete state Markov, or state-dependent, models have been used extensively to probe the role of ion channel dynamics in generating the excitability of neurons [1], cardiac myocytes [2], and pancreatic beta cells [3,4]. Markov models recapitulate channel dynamics by discretizing behavior into a series of states, with transitions between states governed by rate constants that often vary as a function of membrane potential [5]. These Markov models are then inserted into cellular models to simulate action potential waveforms and frequency-dependent properties [3,6–8]. For many types of ion channels, Markov model topologies describing their kinetic and voltage-dependent properties have evolved to reflect refined knowledge of channel behavior and functioning from decades of experimentation. For example, experiments have revealed multiple activation gates and inactivation states of voltage-gated ion channels whose occupancy spans many time domains. States have been continuously added to existing Markov models to improve their ability to account for this additional complexity of the dynamics observed in single channel and macroscopic current experiments [9–13]. For the voltage-gated cardiac Na$^+$ channel, for example, current Markov models reflect multiple stages of channel activation, deactivation, and inactivation from both closed and open states [14,15].

There is a rich history in modeling macroscopic and single channel currents that takes advantage of various topologies, or structures, of Markov models to reflect our understanding of channel dynamics [1,13,16–22]. There have been also numerous studies on parameter identifiability and equivalence [18,23–28]. Both cases, however, have explored a limited collection of topologies either for understanding the details of channel gating or recapitulating general channel dynamics. In 2009, Menon and colleagues surpassed previous efforts by exploring many model topologies through a genetic algorithm that theoretically optimizes model structure in addition to rate parameters. Making random perturbations to optimize model structure, however, is challenging from an optimization point of view because the addition or removal of a state causes a large jump in the parameter landscape. By enumerating the unique channel Markov model topology search space, however, optimization may focus on rate parameterization of these unique structures that thoroughly cover the search space. Enumeration also allows for absolute ranking of structures in order of increasing complexity, so that through examining the performance of multiple topologies, one may estimate the complexity needed to recapitulate that specific dataset.

Systematically identifying various model structures is especially helpful given the range of goals in channel kinetic modelling [29]. A model, for example, may need to reflect new structural information, new functional role(s) of channel interacting proteins [30–32] or, as in the CiPA initiative [33], massive amounts of electrophysiological data to simulate the proarrhythmic effects of drugs. These types of studies may require a model that recapitulates molecular level detail precisely, i.e., for gating studies. Other types of studies, however, may only need a model that captures the principal dynamics of the channels, for example, in simulations of action potentials. By enumerating all possible model structures, we can suggest multiple structural candidates at different levels of complexity for validation of various types of data and complexities of datasets.

While human intuition has laid a solid foundation for early ion channel models [34], there is a great need for a systematic, efficient method to identify possible Markov topologies given a specific experimental dataset. We present an investigation into Markov model topologies examined incrementally in increasing complexity with two voltage-clamp datasets derived from analyses of cardiac fast transient outward ($I_{to,f}$) potassium currents and rapidly activating and inactivating sodium ($I_{Na}$) currents. Multiple topologies invite opportunities to understand how discretized states and rate constants come together to form a successful model of channel

dynamics. This strategy also provides the opportunity to summarize topological features that work well for creating channel models.

## Results

### Identification and enumeration of unique topologies in increasing complexity

Our initial aim was to count how many different topologies are possible for a Markov model with a given number of states. To accomplish this goal, we assumed a Markov model topology with one state designated as open for simulating current with the rest not strictly labeled as in Menon et al. [16]. In terms of graph theory, this open state is called the root. By starting with the root (open state), we could then iteratively evaluate the connectivity of the other states [35,36]. A challenge arose, however, because topologies may appear to be unique by their numbering, even though the state labels are permutations. Thus, our counting algorithm needed to assess whether models were oriented uniquely, as opposed to simply being labeled differently. Using the 3-state topology space as an example, 36 permutations of single rooted topologies are possible (**Fig 1A**). For clarity, **Fig 1B** depicts the three topologies that are unique with respect to the root. A unique graph guarantees that the root is oriented distinctly with respect to the other states. To generate this unique space for greater than three states, topologies of various sizes were tested for isomorphism [37] and only the unique topologies were retained [38]. Parsing from single rooted topology permutations (36) to uniquely oriented state topologies (3) is depicted in **Fig 1C** along with the results of a similar analysis for topologies up to 10 states.

As can be seen in the **Fig 1E**, this parsing dramatically reduced the model search space by orders of magnitude as the number of states increased, providing an upper bound on the topology search problem. However, the number of unique topologies was still on the order of millions for 9 and 10 states, and many topologies cannot plausibly be studied given current constraints on computational resources. Thus, we sought to reduce the number of topologies to be evaluated further by focusing on those that might be the most useful for modeling native ion channel behavior. It is worth noting, however, that future efforts to explore the excluded models further might be feasible and appropriate as computational resources continue to increase.

### Biophysically inspired parsing of unique topologies

To further reduce the number of topologies, the degree of a state, defined as the number of edges (connections) possible to other states given residency in a certain state, must be limited. To accomplish this, we placed a moderate restriction of a maximum of a degree of four on a state (preventing one state from accessing many others) (**S1A Fig**). A state with a high degree implies that a given conformation of the channel has direct access to many different adjacent conformations, each with an associated rate.

Large cycles also introduce additional challenges as the topologies start to represent long-range connections. In other words, states that are far apart in the ion channel excitatory cycle, such as "deep" (or especially stable) inactivated or closed states, may be connected directly. Experiments suggest, however, that a sequence of distinct channel energetic conformations likely take place between these stable states [39,40]. By retaining these long-range connections or cycles, the topology implies that distinct pathways may be bypassed. Like the maximum state connections, therefore, we set a moderate restriction of four on the maximum cycle length to create the focused model search space (**S1B Fig**). Together with the elimination of isomorphic topologies, focusing the search on topologies that meet both biophysical restrictions (**S1C Fig**) reduced the original space for a 10-state model from $10^{14}$ to $10^5$.

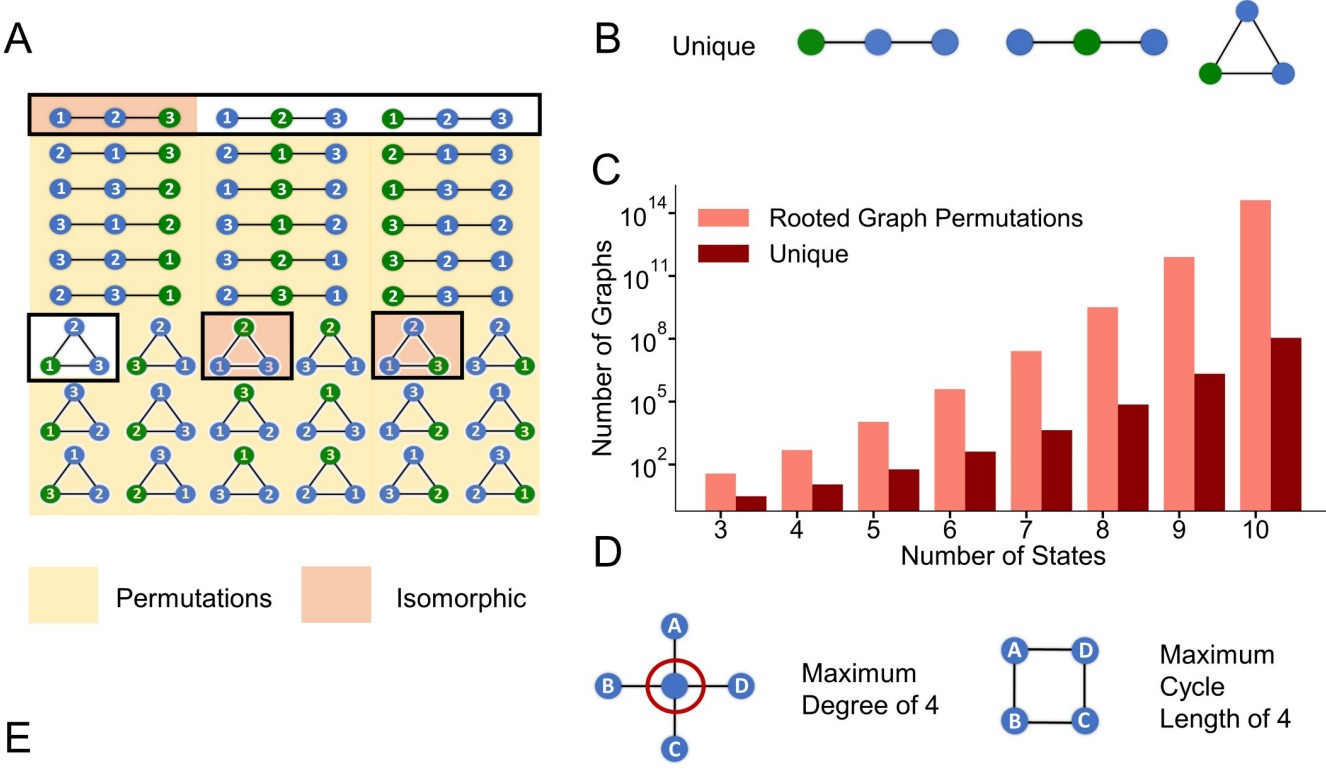

**Fig 1. Reduction in model search space by enumerating unique topologies as a function of the states. A)** Blue states are non-open while the open state (root) is colored green. All 36 permutations of three state rooted topologies are depicted. Permutations outlined in black represent the original six possible rooted topologies of three states. Orange shading represents the three isomorphic permutations of the six possible rooted topologies while the three unshaded topologies correspond to the unique rooted topologies as in **B**. The yellow shaded topologies are the remaining 30 permutations. A reduction from 36 rooted graph permutations to three unique topologies is depicted in **C. C)** Results of a similar graphical enumeration analysis for rooted topologies with 4+ states. **D)** Biophysically inspired restriction of the maximum degree in a graph to 4. Applying this restriction to the unique topologies results in a reduction in a model search space as enumerated in the table. After further restricting the maximum cycle length in a graph to size 4 after the degree restrictions, final graph counts are displayed in **E** as function of the number of states. **E)** Enumeration summary table of rooted graph permutations, rooted topologies, unique rooted topologies, and biophysical restrictions as a function of the number of states.

## Evaluation of unique model topologies

The unique models with varying number of states (**Fig 1**) were sorted according to increasing numbers of free rate constants, proportional to the sum of the number of states and edges (connections) as a measure of model complexity [16]. Two canonical ion channel datasets were utilized to identify possible model structures needed to recapitulate channel dynamics:

the voltage-gated, rapidly activating and inactivating, cardiac sodium current ($I_{Na}$) and the voltage-gated, fast transient outward, potassium current ($I_{to,f}$). In mammalian cardiac myocytes, $I_{Na}$ is responsible for the upstroke of the action potential [41], while $I_{to,f}$ contributes to early repolarization and, notably, is responsible for generating the early "notch" of the action potential that is prominent in epicardial ventricular myocytes in many large mammals, including humans [42].

## Rate parameter optimization

To evaluate the suitability of a particular topology to serve as a Markov model, rate parameters needed to be optimized to capture the trends in the voltage-clamp data. As described previously in Menon et al.[16] and Teed et al.[20], rate constants are of the form:

$$r_{ij} = \exp\left(a + \left(b * tanh\left(\frac{v + args_1}{args_2}\right)\right)\right) \tag{1}$$

where $r_{ij}$ is the rate from state $j$ to state $i$, $a$ and $b$ are optimized parameters, $v$ is voltage in (mV), $args_1$ and $args_2$ are optimized parameters as part of the sigmoidal voltage function [20]. Rate parameters were optimized using simulated annealing with adaptive temperature control [43] (Eqs 2–5). To minimize the dependence of optimal rate parameters of a unique graph on initial optimization conditions, multiple starts of simulated annealing were performed with initial rate parameters scaled according to a quasi-random Sobol sequence [44] (Eq 9) to thoroughly explore the parameter space.

## Overfitting prevention

An important goal was to find rate parameters that captured the trends in the voltage-clamp data while also avoiding overfitting. Specifically, overfitting occurs when the model fails to represent the general trends in the data owing to focusing on fitting all experimental data perfectly [45]. Our aim was to maximize the chance the Markov models would successfully predict channel dynamics that were not necessarily included in our canonical datasets (maximize generalizability). Further, we also wanted to quantify how likely it was that overfitting was occurring during the optimization process, and when, so that the process could be terminated. Our rate parameter optimization routine included a measure to halt optimization if likely overfitting occurs using methods borrowed from training neural networks [46]. To accomplish this aim, experimental data were split into training and validation sets.

The trajectory of the reduction in training cost (progress) was tracked periodically throughout the optimization along with the current validation set cost with respect to the minimum (generalization loss). **S2A Fig** shows representative training and validation cost trajectories during a model optimization. As is evident, the training cost slowly, but steadily, decreases throughout the optimization, while the validation cost varies erratically. If the validation cost first decreased but then consistently monotonically increased after some point in the optimization, there would be a clear stopping point to prevent overfitting. However, the spiking in the validation cost trajectory makes it difficult to identify a clear stopping point. To quantify where overfitting is likely occurring and, therefore, where to terminate the optimization, measurements of progress and generalization loss were computed at regular epochs (**S2B Fig**). These measures give insight into the improvement in the fit of the training data in comparison to the validation set. As described previously [46], using this ratio, there are many strategies that can be used to determine when to halt the optimization. Here, we used the condition that three consecutive increases in the overfitting ratio results in early termination. The trajectories of the progress and generalization loss measures demonstrate (**S2B Fig**) that this ratio allows

enough flexibility for the validation cost to fluctuate throughout the optimization until a validation cost minimum is reached with termination shortly thereafter.

## Reduction of model solution stiffness

We also considered minimizing model solution stiffness while optimizing rate parameters for a given unique topology. Systems of differential equations are considered "stiff" when the derivatives of the function are large near the solution, requiring very small time steps to be taken for solution stability [47]. Implicit differential equation solvers can be utilized to solve stiff systems more efficiently, although these are more computationally demanding than explicit solvers and often require specification of the Jacobian. The long-term goal here is to be able to incorporate the optimized channel Markov models into cellular and tissue models of membrane excitability. When scaling up to the cell or tissue level, a given ion channel model may be solved thousands and thousands of times. Thus, it is crucial that computational solving time is considered when creating the individual ion channel models. To quantify the stiffness of each model solution, the condition numbers of the transition matrices were estimated at various membrane voltages [48,49] (Eq 7). Estimated reciprocal condition numbers larger than a threshold were averaged and proportionally contributed to the model cost as the stiffness penalty (Eq 8).

## Application to the human ventricular $I_{to,f}$ dataset

We first present data from multiple optimizations runs with the overfitting and stiffness penalties for the human ventricular $I_{to,f}$ dataset. **Fig 2A** displays the frequencies of optimization iterations completed as a function of increasing free rate constants, while optimizing with overfitting and stiffness penalties. Iterations completed from multiple optimizations with differing starting conditions of each model are displayed. The relative weights of the black dots represent the frequency of maximum optimizations completed. As the number of free rate constants increases, most optimization runs reach the maximum number of allowed iterations (large black clusters). However, the intensity of the trailing black dots, which represent the number of optimization iterations completed before being terminated early, increases as well. This result supports the notion that overfitting becomes more problematic as the model complexity increases [24]. **Fig 2B** depicts the associated normalized costs for the model populations after the optimization iterations specified in **Fig 2A**. The large spread in the normalized costs illustrates the importance of running optimizations multiple times when estimating the absolute minimum cost of models with varying complexities. Of note, extremely high normalized costs result from especially poor optimization starting conditions.

Focusing just on the absolute minimum costs seen across the model populations reveals that the minimum cost trends downward as complexity increases: models with six and seven free rate constants produce minimum costs on the order of 10 times less than models with four free rate constants. Tracking this absolute minimum cost as the number of free rate constants increases reveals a point of diminishing returns. The absolute minimum cost decreases appreciably when comparing model populations with four, five, and six free rate constants. However, there is hardly any change when comparing models with six and seven free rate constants. **S3 Fig** displays the various stiffness penalties as a function of model complexity. The smallest stiffness penalty seen across all optimization starts leveling out for model topologies with six and seven free rate constants, as does normalized cost. This point of diminishing returns in absolute minimum cost and stiffness penalties at six free rate constants suggests that this complexity may be optimal to model the canonical dynamics of $I_{to,f}$ in this specific dataset. Models with six to seven free rate constants have the potential to prioritize good fidelity fits to

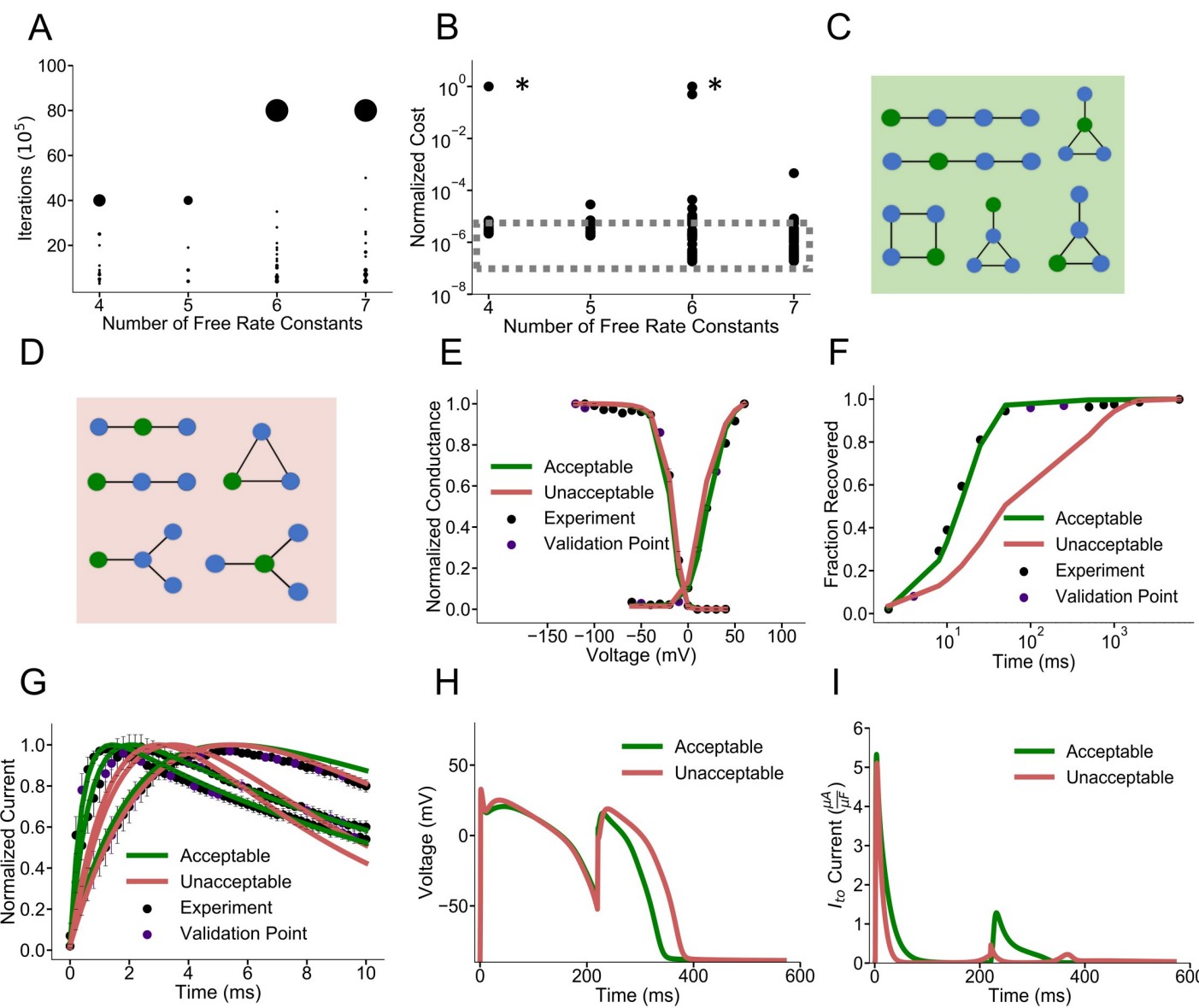

**Fig 2. Identification of possible structures for the I$_{to,f}$ dataset. A)** Optimization iterations completed as a function of free rate constants. Weighting of dots represent the frequency of models completing a specified number of iterations. Most models complete the generous maximum iteration limit (40,000,000) with 4 and 5 free rate parameters while few models complete less iterations due to early stopping from the overfitting criterion. Models with 6 and 7 free rate parameters may run for longer (maximum 80,000,000 iterations displayed) while a greater fraction of models are terminated early due to overfitting criterion. Data points include the multiple starting conditions for each model to reduce the dependence of the minimum solution on initial conditions **B)** Distribution of normalized costs for each model with multiple starts after completing optimization iterations as depicted in **A**. Especially bad Sobol starting conditions are asterisked. The absolute minimum costs are outlined in the dashed grey box. The point of diminishing returns is at 6 and 7 free rate constants. **C)** Topologies producing acceptable fits and **D)** unacceptable fits for I$_{to,f}$. Acceptable fits include models with four states and sufficient connections to separate the activation and inactivation domains. **E-G)** Representative models fits for steady state activation, inactivation, recovery from inactivation and current traces for models in the acceptable and unacceptable model categories. Unacceptable models generally have very slow recovery from inactivation. **H)** Simulated action potentials with representative acceptable and unacceptable I$_{to,f}$ currents under a S1-S2 protocol. **I)** Corresponding simulated I$_{to,f}$ currents under the S1-S2 protocol in **A**. Acceptable currents (green) and unacceptable currents (red) most differ in magnitude at around 200 ms into the S1 action potential with much less unacceptable I$_{to,f}$ currents. This corresponds with the slow recovery from inactivation depicted in **F)** where at ~200 ms the acceptable models have fully recovered while unacceptable models are only half recovered.

voltage protocols while minimizing overfitting potential. S4 **Fig** displays the topology of an example model for the I$_{to,f}$ dataset with six free rate constants along the fits to the voltage protocols.

## Classification of acceptable and unacceptable possible $I_{to,f}$ model models

Our next aim was to categorize all $I_{to,f}$ model topologies as acceptable or unacceptable, based on the minimum cost seen across all optimization starts. A model with a minimum cost no greater than 300% of the absolute minimum was deemed "acceptable." These topologies are displayed in **Fig 2C.** Models that consistently produced poor voltage protocol fits are displayed in **Fig 2D**. These unacceptable models have four to five free rate constants and consistently produce higher normalized cost values, as illustrated in **Fig 2B**. Representative model fits from the acceptable (green) and unacceptable (red) model categories to the voltage-clamp protocols are displayed in **Fig 2E-G**. Unacceptable models tended to produce fits with slow recovery from inactivation (**Fig 2F**) with little impact on the other protocols (**Fig 2E and 2G**). **S5 Fig** tracks state occupancy as function of time during the recovery from inactivation protocol. Acceptable models have enough complexity to recapitulate the slow timescale of steady state and the faster timescale of recovery at -70 mV. Unacceptable models show slow recovery from inactivation because the rates cannot be sufficiently fast during recovery while also fitting steady state conditions.

## Validation of $I_{to,f}$ dataset cost threshold

The modeled $I_{to,f}$ was included into a human ventricular myocyte action potential model [50] under a S1-S2 pulse protocol that simulates repetitive excitation (see Methods) to validate our categorization of acceptable and unacceptable models based on cost. An S2 stimulus given at around 200 ms into the S1 action potential revealed that unacceptable models can lead to a longer action potential duration (**Fig 2H**). Analyzing the corresponding currents revealed that, at 200 ms into the S1 action potential, the magnitude of $I_{to,f}$ generated was much lower in the unacceptable model compared to the representative acceptable modeled $I_{to,f}$. (**Fig 2I**). This result is in accordance with the lagging fraction of recovered channels for the unacceptable models at 200 ms as depicted in **Fig 2F**. As the magnitude of $I_{to,f}$ influences the notch and plateau potentials, which will secondarily impact calcium entry and excitation-contraction coupling [51], the deficiencies in the simple models could result in inaccurate cellular and tissue level predictions. Analyzing the modeled currents under this S1-S2 protocol to stimulate the impact of changing heart rate reveals the precise window of time over which the inactivation and incomplete recovery from inactivation of $I_{to,f}$ channels could manifest itself at the cellular level. The overly simplistic models, with complexities below 6–7 free rate constants, fail to capture the full dynamics of $I_{to,f}$ when simulating rate-dependent effects on action potential waveforms.

## Application to the mouse atrial myocyte $I_{Na}$ dataset

Subsequent efforts were aimed at repeating the strategy outlined above on an available (mouse atrial myocyte) $I_{Na}$ dataset to find possible models for a channel with more complex dynamics and explore more complex topologies. The cardiac sodium current is more complex than $I_{to,f}$ because of the multiple time scales involved with its activation, inactivation and recovery from inactivation [41]. As with the modeling of $I_{to,f}$ however, the initial goal was to sort the studied model topologies in order of increasing complexity into the acceptable and unacceptable model categories based on cost (**S6 and S7 Figs**). Although examining the model fits to the individual voltage protocols helped classify acceptable and unacceptable topologies for the $I_{to,f}$ dataset, acceptable and unacceptable $I_{Na}$ models did not display a severe protocol fitting deficiency, i.e., none of the voltage-clamp protocol fits were consistently poor (**S6E–S6G Fig**). Thus, when validating the simulated acceptable and unacceptable $I_{Na}$ currents in the action

potential [52], there were no appreciable differences in morphology (**S6H Fig**), because the models were being primarily separated based on the stiffness penalty.

## Application to HEK-293 $I_{Na}$ dataset

Because the mouse atrial myocyte $I_{Na}$ acceptable and unacceptable models could not be distinguished by the representative individual protocol fits, we repeated the analysis of $I_{Na}$ using a previously published $I_{Na}$ dataset, generated for heterologously expressed *SCN5A-* (Nav1.5-) encoded) $I_{Na}$ in HEK-293 cells, that includes slow and intermediate components of recovery from inactivation [53–55]. We predicted that a more complicated recovery from inactivation protocol would require greater model complexity to fit all voltage protocols, so that topologies with few free rate constants would fail to reproduce all kinetics. **Fig 3A and 3B** show representative acceptable and unacceptable model topologies after rate parameter optimization, confirming this hypothesis. At least eight free rate constants were required in the models to fit this complex dataset (**Fig 3A**)**,** while sparsely connected topologies and those with fewer than eight free rate constants (**Fig 3B**) failed to reproduce all protocols with good fidelity. Representative unacceptable and acceptable model fits to individual voltage protocols are shown in **Fig 3C–3F**. **Fig 3E** depicts the more complex protocol of recovery from use dependent block (RUDB). The repetitive voltage steps generating RUDB (see Methods for protocol) allows for the slower and intermediate components of recovery from inactivation of the $I_{Na}$ models to be parameterized. Unacceptable models show slow recovery from fast inactivation (**Fig 3D**), poor model fidelity to the intermediate and slow timescales of recovery from inactivation under RUDB (**Fig 3E**), and poor voltage-dependent inactivation kinetics (**Fig 3F**). This result suggests that it is quite difficult to parameterize simpler models to capture various types of recovery from inactivation accurately, along with other voltage protocols.

## Validation of the acceptable and unacceptable $I_{Na}$ HEK-293 dataset cost threshold

To validate the HEK $I_{Na}$ dataset categorization of acceptable and unacceptable models based on cost, models were first included into a single cell ventricular action potential model[52]. Incorporation of these modeled currents shows that unacceptable $I_{Na}$ HEK models tend to produce action potentials that fail to repolarize, while representative acceptable models successfully repolarize (**Fig 3G**)**.** Plotting the simulated $I_{Na}$ reveals that representative unacceptable models have abnormal gating into the action potential (late component) that hinders action potential repolarization (**Fig 3H**).

## Performance summary of all model solutions: $I_{to,f}$

Up until this point, we reported the minimum cost seen across all optimization starts when describing a model as acceptable or unacceptable. In **Fig 4**, we report the end performance of each optimization start (20 displayed) for each topology in addition to all low-cost model solutions in the reported optimization history. This presentation reflects the diversity in rate parameter parameterizations (i.e. model solutions) during and after multiple optimizations. For example, for a given topology there may be vastly different sets of parameters that produce a model with an acceptable cost. **Fig 4A** summarizes the performance of all studied $I_{to,f}$ model topologies to serve as Markov models when sorted by number of states and connections. Acceptable and unacceptable labels correspond to the action potential validation performance as depicted in **Fig 2H**. However, 23 out of 156 low cost model solutions on topologies depicted in **Fig 2C,** failed to recapitulate the increased $I_{to,f}$ evident using the S1-S2 protocol. We labeled those model solutions as "tentative" to reflect the fact that, despite good fits to the

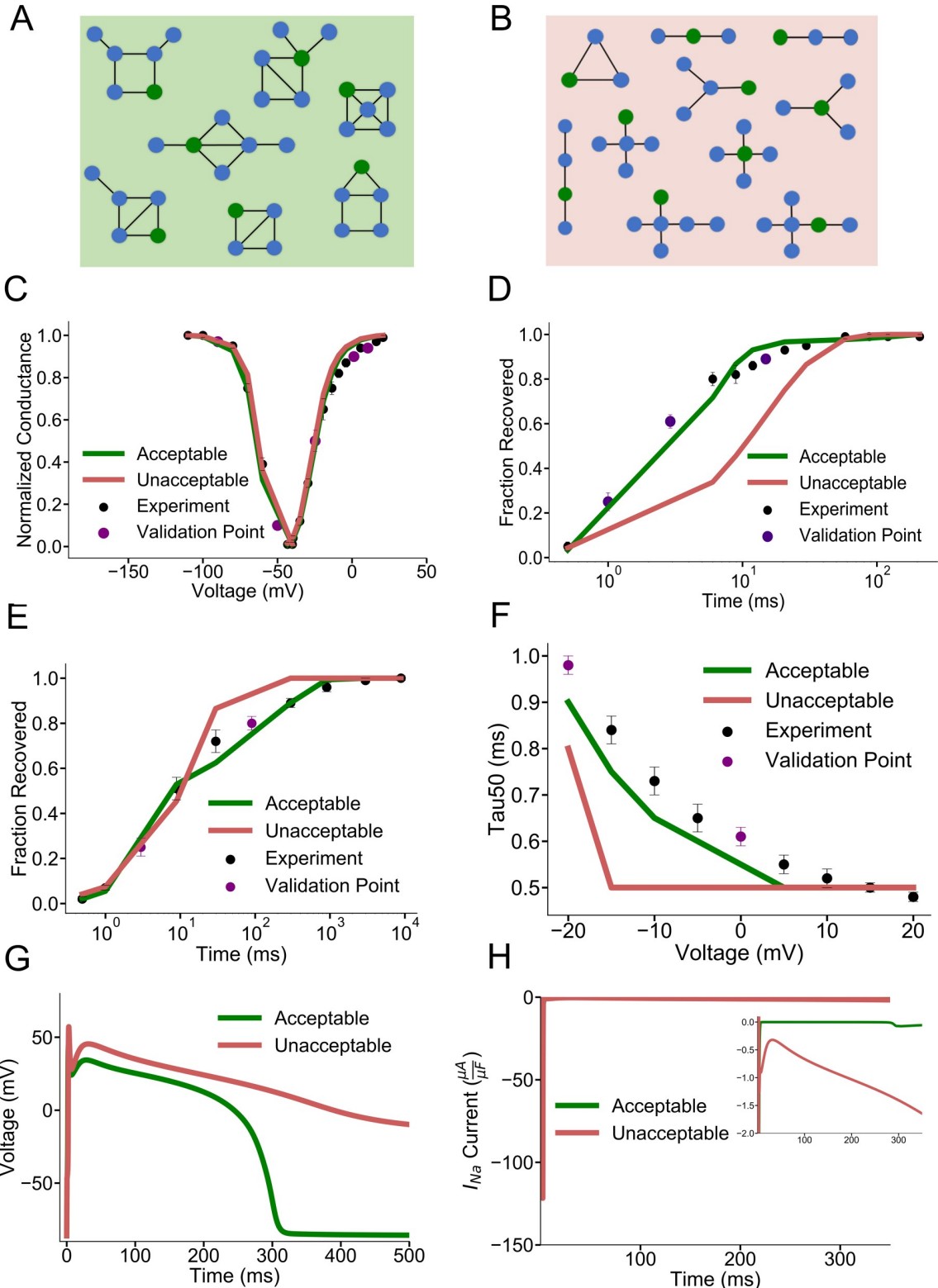

**Fig 3. Identification of possible model structures for the $I_{Na}$ HEK dataset A)** Representative acceptable model topologies of the more complex HEK $I_{Na}$ dataset. Topologies have at least 8 free rate constants. **B)** Representative unacceptable model topologies. Topologies have fewer than 8 free rate constants or are sparsely connected. **C)** Representative acceptable and unacceptable model fits to steady state activation and inactivation **D)** Representative fits to fast recovery from inactivation **E)** Representative fits to recovery

from use dependent block which includes timescales of fast, intermediate, and slow recovery from inactivation. **F)** Representative model fits to the time constant of 50% inactivation of the peak sodium current **G)** Simulated action potentials with representative acceptable and unacceptable $I_{Na}$ modeled currents. Unacceptable models have varying degrees of late $I_{Na}$ current) **H)** Corresponding representative acceptable and unacceptable $I_{Na}$ currents during the action potentials in **G** with a magnified inset highlighting the late $I_{Na}$ current.

voltage-clamp protocols, these models failed to perform like the other acceptable models in the action potential simulations. In **Fig 4A**, 85% of four-state low-cost model solutions with three or four edges were labeled as acceptable compared to 15% tentative. **Fig 4B** shows that low-cost topologies with lower root degree (fewer open state connections) tend to be more successful. For topologies with three edges, all topologies are labeled acceptable with a root degree of one while all topologies with a root degree of two are labeled tentative. For topologies with four edges, all topologies are acceptable with a root degree of one, 80% are acceptable with root degree of two, and 50% are acceptable with root degree of three. These observations are expected given that topologies with lower root degrees with few states result in a "spoke" layout. These "spoke" topologies with a central open state require careful rate parameterization to prevent "bursting" of the channel over time at various membrane potentials. This translates into a more difficult optimization problem, and so the optimizer struggles to find a satisfactory solution given the finite optimization limits set.

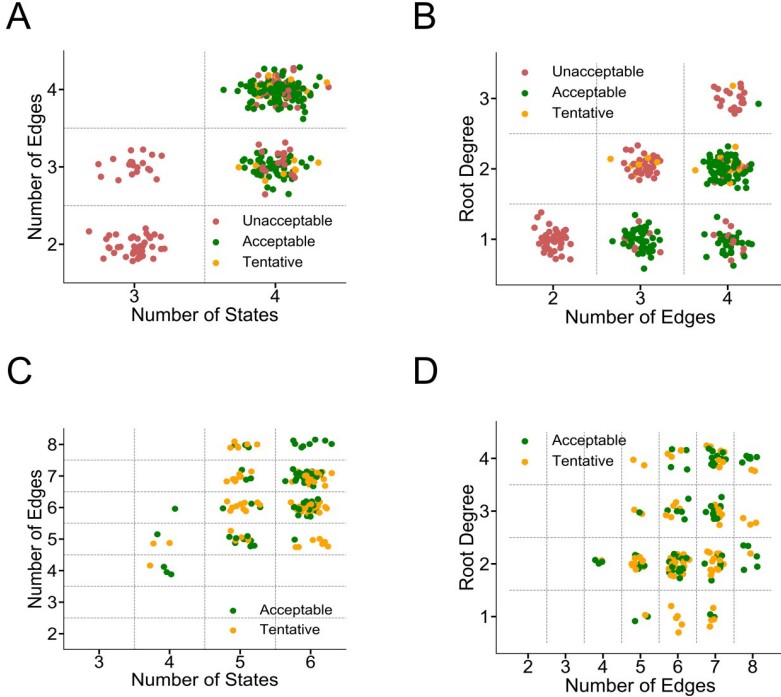

**Fig 4. Summary of Markov model performance for all topologies studied. A)** and **B)** Performance summary of all $I_{to,f}$ topologies studied. Unacceptable and acceptable labels are as previously defined. Tentative yellow topologies produced acceptable voltage protocol fits but did not perform as other acceptable models in the action potential validation. In **A)** more states help topologies find tentative and acceptable solutions. **B)** Topologies with lower root degrees (open state connections) tend to create acceptable models **C** and **D)** Performance summary of all $I_{Na}$ topologies studied. Unacceptable and acceptable labels are as previously defined. Tentative yellow topologies produced acceptable voltage protocol fits but did not perform like other acceptable models in the action potential validation (different degrees of repolarization failure). **C)** Topologies with more than the minimum number of edges tend to yield more tentative or acceptable Markov models. **D)** As edges increase, topologies most benefit from higher root degrees (more open state connections) to serve as successful Markov models. Panels C and D do not show unacceptable models for clarity.

## Performance summary of all model solutions: $I_{Na}$ HEK

**Fig 4C and 4D** summarize the performance of all model topologies studied across all optimization starts and history when training on the HEK $I_{Na}$ dataset. Acceptable labels correspond to the action potential validation performance as depicted in **Fig 3G**. However, 89 out of 169 low cost model solutions on topologies depicted in **Fig 3B,** failed to perform as other acceptable model solutions in a single cellular ventricular action potential or in reported conduction velocities in a 100 cell cable [52] (**Fig 5B**). As before with the $I_{to,f}$ dataset, these solutions are labelled as tentative. For clarity, only the acceptable and tentative models are displayed in **Fig 4C and 4D**, whereas **S9 Fig** also includes unacceptable models. When sorting based on numbers of states and edges, topologies with more than the minimum number of edges tend to serve as successful Markov models (**Fig 4C**). In other words, sparsely connected topologies do not have the complexity to recapitulate channel dynamics. Among four, five, and six state models with more than the minimum number of edges, about half of all low-cost solutions were ultimately labelled acceptable. Topologies with the minimum number of edges had maximally 13% acceptable. When sorting based on total topology connections versus root degree (**Fig 4D**), higher root degrees generally aide in creating successful Markov models as the number of edges in a topology increases. For example, for models with root degrees of four, between five edges and eight edges, the acceptable labelling rate increases from 0% to 75%.

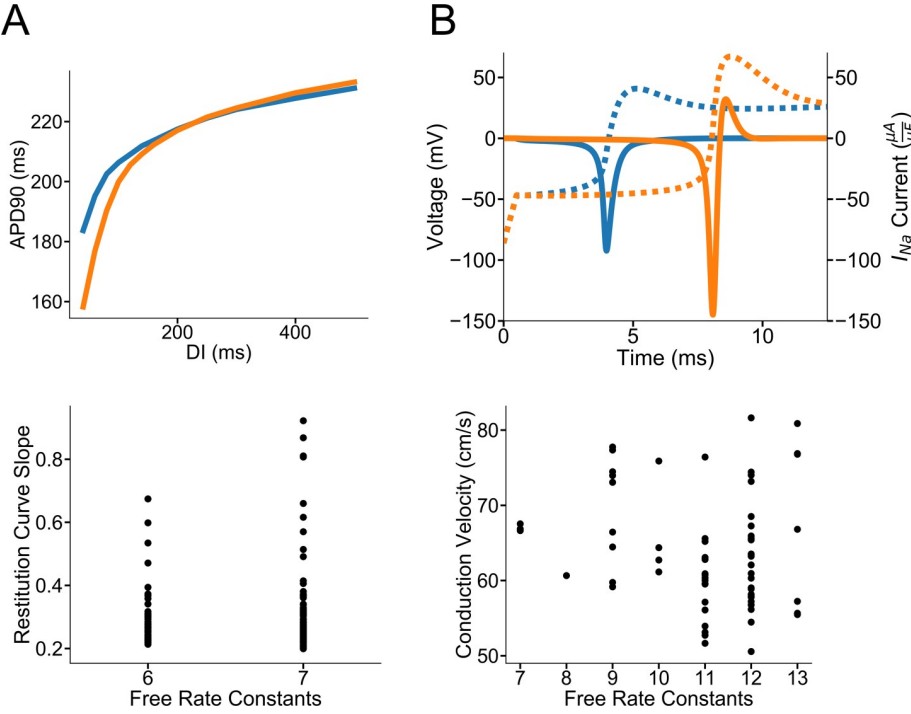

**Fig 5. Validation of acceptable model solutions in cellular and tissue simulations.** Model solutions show diverse behavior in cellular and tissue simulations despite good fidelity fits to the training voltage protocols. **A)** Representative shallow and steep slopes of acceptable $I_{to,f}$ models in a dynamic restitution protocol. Slopes are calculated from DI intervals between 60 and 80 ms. Most acceptable model solutions have shallower slopes between 0.2–0.4. There are 12 acceptable model solutions that have slopes higher than 0.4 indicating larger $I_{to,f}$ currents (see **S10 Fig**). **B)** Conduction velocity variability results from variability in the precise $I_{Na}$ gating. Despite low-cost protocol fits (**S11 Fig**), the activation of the $I_{Na}$ current varies between 4 and 8 ms with conduction velocities of 67 (blue) and 56 (orange) cm/s, respectively for two representative acceptable models. Dashed lines represent the corresponding action potentials upstrokes. Plotted below are all acceptable $I_{Na}$ model conduction velocities in a 100-cell ten Tusscher cable. Reported conduction velocities are from Cell 50.

Once a topology has seven connections, however, the topology likely has six states, so open state connections may still be incorporated in a cycle. This arrangement allows the topology to recapitulate more complex dynamics. Thus, many open state connections are not automatically detrimental for topologies with more complexity.

## Acceptable solution variability

When comparing all acceptable $I_{to,f}$ model solutions in a dynamic restitution protocol (outside of the specific S1-S2 described above), there is considerable variability in the slopes of the restitution curves between DI intervals of 60 to 80 ms (**Fig 5A**). Of the 133 acceptable, action potential-validated solutions, there are 12 solutions with restitutions slopes greater than 0.4. The rest of the 121 solutions have slopes between 0.2–0.4. This variability stems not from the model complexity, but rather from differences in $I_{to,f}$ inactivation gating after the $10^{th}$ S1 stimulus. **S10 Fig** shows representative model voltage and $I_{to,f}$ current traces with shallower and steeper restitution slopes during the protocol. Despite both model solutions having good training voltage protocol fits and recapitulating in the specific S1-S2 $I_{to,f}$ kinetics, $I_{to,f}$ inactivation is quite different between acceptable solutions. These representative models first illustrate that seemingly similar acceptable $I_{to,f}$ models result in differences on the restitution properties of cardiac myocytes despite the fact that $I_{to,f}$ is only one of many repolarizing potassium currents. Most importantly, even with protocol fits that are both in the acceptable range of cost, the restitution kinetics at DI intervals at 60–80 ms are inconsistently constrained.

When incorporating acceptable $I_{Na}$ HEK models in a 100-cell cable, there is considerable variation in the conduction velocities that does not depend on model complexity. Representative acceptable action potential validated models (**S11 Fig**) are shown in **Fig 5B** with differences in the onset of $I_{Na}$ activation at 4 ms and 8 ms resulting in conduction velocities of 67 and 56 cm/s, respectively. Of all the 69 acceptable model solutions plotted in **Fig 5B**, most fall between 50–75 cm/s with 8 acceptable solutions above 75 cm/s. Solutions vary because of altered activation timing compared to the original sodium current in the ventricular action potential model [52]. **S12 Fig** further illustrates that the conduction velocity does not depend on inactivation either when controlling for the extent of activation. These findings combined underscore the importance of having access to original current traces, rather than only to measured/reported kinetic time constants, for training robust channel models with less parameter unidentifiability, as previously described by Mirams and colleagues [28].

## Successful Markov model topological features

Taken together, sparsely connected topologies tend not to serve as successful Markov models. When topologies have five states or less, more open state connections result in harder to parameterize "spoke" topologies. However, when topologies become complex enough (many connections or states), this is not an inherent detriment to the topology. A root then may have many connections to the rest of the model and still be incorporated in a cycle, for example, so the topology has inherently more capacity to represent more complexity. This thinking suggests potential filters to further parse the model space beyond degree and cycle length restrictions. For topologies with three, four, or five, states (2–3) open state connections should be emphasized while larger models will benefit from a variety of open state connections. Given limited computational resources, topologies with more than the minimum number of connections should be prioritized. For example, in the seven-state model space, there are 1483 topologies after applying restrictions on state connections and cycle length. (**Fig 1E**). If topologies with minimal connections (6–7) were excluded, another 166 topologies could be further eliminated.

## Discussion and Future Directions

We present a robust, systematic method to identify all possible model topologies for simulating ionic currents, given experimental datasets of canonical ion channel dynamics. The routine moves through various topologies in a stepwise fashion as a function of the number of free rate constants. By examining the diminishing returns in model cost, one may visualize the complexity of various topologies to best balance between model solution fidelity, overfitting [24], and stiffness for the specific experimental dataset. Depending on the goals of the kinetic modeling study undertaken, the user may still wish to utilize a topology with more or less complexity identified in this routine. In these cases, we provide an organized framework for users to validate topologies for their specific goals. We also demonstrate the robustness of this methodology by modeling two voltage-gated currents: fast transient outward ($I_{to,f}$) potassium currents and rapidly activating and inactivating sodium ($I_{Na}$) currents.

Using the formulation of the single tracked open state as in Menon et al. [16], we were able to enumerate the model topology search space. This exhaustive enumeration ensures complete coverage of the model search space, rather than relying on random perturbations or limited collections of various model topologies during optimization. As illustrated in **Fig 1**, this enumeration is crucial when working with topologies with six states and greater because of permutations. Enumeration also allows for topologies to be evaluated systematically in terms of model complexity, which is critical for identifying the amount of complexity available in possible structures for an experimental dataset. Through restrictions on state connections and elimination of long-range connections, we were able to further parse this model search space given decades of biophysical experimental insight. The results here further suggest an additional filter that eliminates topologies that are sparsely connected. It may also prove fruitful to emphasize topologies with lower root degrees (fewer open state connections) when working a few states but higher root degrees (many open state connections) (up to four) in topologies with seven states and higher.

Optimizing models with greater numbers of parameters does not easily lead to a "perfect" model with a normalized cost of zero. With more computationally expensive starts and larger maximum simulation iterations without overfitting prevention, we may expect to see the cost function decrease to zero given infinite optimization time. However, given constraints on time and computing resources, we present the best costs at least 100,000 iterations beyond a change of 20% or less in cost (unless terminated early for overfitting). We see a point of diminishing returns in the normalized cost function and suggest optimal complexity based on the dataset.

## Features of successful Markov models

Grouping acceptable and unacceptable models of each dataset allows one to begin to explore the question of what makes a suitable topology for Markov models of ion channel dynamics. The $I_{to,f}$ dataset showed that three-state models are not sufficiently complex to recapitulate dynamics, but four-state topologies without cycles can do so successfully. The example minimal four-state linear model for $I_{to,f}$ illustrated in **S4A** and **S5C** **Figs** does not contain a cycle between open, hypothetically closed and inactivated states, as is commonly used to model voltage-gated ion channels [14,15]. Ion channel Markov models need not always conform to human intuition of the underlying structural mechanisms to reproduce a dataset with good fidelity. A modeler has great power in determining how much complexity is needed for the computational problem at hand [56]. More complex models, for example, may be appropriate in studies focused on exploring channel gating precisely. In studies that require attention to fast computations at the tissue level, a simpler, less stiff model, may be the most appropriate.

The example models of the simpler $I_{Na}$ and the more complex $I_{Na}$ datasets are more consistent with mechanistic intuition derived from consideration of experimental data. The example

topology for $I_{Na}$ in **S8A Fig**, for example, contains a cycle, which is consistent with prior cyclic models of ion channel excitability with connections between activated, inactivated, and closed states (14,15). Acceptable models depicted in **Figs 3A and S6C** contain at least one cycle. Unacceptable $I_{Na}$ models tend to be sparsely connected, thus not able to include cycles (aside from the three-state cyclic model), which leads to insufficient complexity to model fast channel dynamics. While single-channel models are successful with sparse connections, fast macroscopic kinetics may necessitate a cycle to prevent channel "bursting" through the designated open state during simulation. Performing analyses of the state probabilities during voltage protocols after using our procedure allows one to gain insight into the mechanisms of the model and discover surprising topologies that successfully recapitulate the protocols.

Our computational routine identifies the best model topologies given an experimental dataset, so collections of the best topologies should evolve as the experimental dataset grows. Our approach also provides a framework for researchers to run the routine with datasets of various sizes to identify collections of optimal models for each dataset. With various topologies in hand, one may begin to understand what topological features are emphasized under certain voltage protocols and combinations of protocols.

## Validation of channel models in cardiac action potentials

Validating the cost thresholds of acceptable and unacceptable current models when these are incorporated (with multiple other ionic conductances) into action potential models allows one to begin connecting how fits to voltage-clamp data will result in electrophysiological differences at the cellular level. In most cases, the action potential morphology may be predicted based on the cost of the model. In the case of $I_{to,f}$ the kinetics of recovery of the channels from inactivation alters the early and late phases of the action potential consistent with marked frequency-dependent effects. The poor protocol fits in the HEK $I_{Na}$ dataset, however, commonly resulted in severe action potential repolarization abnormalities. This result might have been expected, given that $I_{Na}$ is solely responsible for the upstroke of the action potential in atrial and ventricular cells, while $I_{to,f}$ is one of many currents responsible for repolarization.

It is also interesting to note that 23 low cost $I_{to,f}$ models (out of 156 acceptable model solutions based on cost) and 89 HEK $I_{Na}$ models (out of 169 acceptable model solutions based on cost), did not perform as well as the other acceptable, low cost models when incorporated into the action potential model. This type of discrepancy is not uncommon in the ion channel modeling field. Ion channel modelers have traditionally specified one model topology to represent a complex dataset. If the optimized rate parameters for the topology failed to produce an action potential, the modeler would blame the topology and adjust the states and connections accordingly. In marked contrast, the approach developed here explores the performance of all possible model topologies. As a result, this discrepancy between low-cost models and consistent, satisfactory behavior in action potential simulations becomes more apparent. The observation that optimized rate parameters can lead to satisfactory training data fits, while also behaving differently in modeling the action potential, highlights the issue of whether a training dataset contains enough information to reliably constrain all model rate parameters consistently. This problem of parameter identifiability has been thoroughly discussed and quantified in ion channel and cellular models of excitability [18,23–28]. This phenomenon is illustrated by the $I_{to,f}$ restitution protocol slopes and the conduction velocity estimates for the acceptable $I_{Na}$ models in **Fig 5**. As illustrated in **S10 Fig**, similar $I_{to,f}$ models can yield currents with different kinetic properties and that differentially affect the shape of the restitution protocol. **S11 and S12 Figs** shows that reasonable fits to the voltage protocols for $I_{Na}$ do not reliability constrain activation, resulting in variability in conduction velocities. As discussed by Mirams et al.

[28], whenever possible, experimentalists should make raw current traces available for training robust channel models in addition to providing derived time constants.

Validating model performance in cellular and tissue action potential simulations serves as a crucial test to detect early signs of parameter unidentifiability and overfitting. Clearly, acceptable fits at the channel level can result in a variety of kinetics and dynamics and the cellular and tissue levels as shown here with $I_{to,f}$ and $I_{Na}$ acceptable models. To differentiate models based on these higher scale behaviors may require additional channel level experimental data or more refined voltage protocols to train model parameters more efficiently and with greater certainty as was previously done with hERG channels [57]. Repeating the analysis with different protocols may further reveal which parameters are most critical for successful action potential generation, depending on the simulation to be performed and those most likely to suffer from parameter unidentifiability. We anticipate this validation will be especially crucial when creating channel models that require greater complexity (more free rate constants) when recapitulating molecular detail, for example, and making arrhythmogenic predictions at the higher scales.

The acceptable model solution totals for both ($I_{to,f}$ and $I_{Na}$) datasets included model solutions based on the costs found at the end of each optimization and at intermediate recorded time points. For $I_{to,f}$, there were 156 acceptable model solutions for the 11 distinct topologies studied. For the HEK $I_{Na}$ dataset, 167 distinct topologies were studied, but only 169 acceptable model solutions were found across all optimizations. Thus, our optimization method found a relative abundance of acceptable solutions for the $I_{to,f}$ dataset compared to the HEK $I_{Na}$ dataset. This difference is likely attributable to the more complex nature of the HEK $I_{Na}$ dataset with the faster kinetics the RUDB protocol, for example, which would require precise parameterization. When tracking the cost over time in the optimization for both datasets, the $I_{to,f}$ optimization proceeded smoothly with a variety of acceptable solutions while the HEK $I_{Na}$ optimization trajectory was substantially more jagged.

## Limitations

There are limitations to the approach presented, and these may be addressed by future work. As depicted in **Figs 2A** and **S6A**, tracking three consecutive increases in the ratio of the generalization loss and progress successfully truncated optimizations, but future studies that explore other validation quantification cutoffs as the optimization problem evolves [24] would be valuable. We use the nonlinear optimization technique of simulated annealing with adaptive temperature control in this study, but more recent algorithms, such as particle swarms [17], may prove to be faster or more accurate. We also anticipate that other additions to the simulated annealing core routine could be helpful, such as adaptive simulated annealing [58] or differential evolution [59] in future studies.

This work provides a framework to identify multiple topology models for canonical ion channel kinetics. By providing open-source code of the computational routine, which could be extended to multiple open states, others may apply this routine to their biophysical systems. These populations of possible model topologies may suggest further experiments for validation of their behavior and may even elucidate more refined voltage-protocols for training the models. They will also allow for connections of the mechanistic underpinnings of the channel through analysis of their topology, rate constants, and state probabilities during voltage protocols. This intuition will prove invaluable when building models based on these topologies in future studies that recapitulate structural and drug interaction data.

## Methods

### Generation of nonisomorphic rooted unlabeled, connected topologies

Nonisomorphic (unique) topologies were generated using nauty (http://users.cecs.anu.edu.au/~bdm/nauty/) a C-based graph isomorphism testing routine. All connected, unlabeled topologies were generated with the specified number of states and with up to the maximum number of edges. (N(N-1))/2. The states in the topologies were then colored in two shades (root and nonrooted) and then imported into a routine to test for isomorphism through canonical labeling[37]. While this study used datasets with a single rooted, open state, the canonical labelling can also be used to detect isomorphisms for topologies with multiple open states. These topologies were then imported and parsed in Python using the NetworkX package based on maximum degree and cycle length. After recording the degrees of all nodes in the graph, topologies were retained if all nodes did not exceed the maximum node degree restriction. Maximum cycle length was used to parse based on long-range connections in graph (see Results for an explanation for long-range connections). The routine 'cycle_basis' was used to identify the cycles in the graph and only topologies with cycles not exceeding the limit were retained.

### Electrophysiological recordings

Voltage-clamp recordings for the simpler $I_{Na}$ dataset were obtained from mouse left atrial myocytes at room (20–22˚C) temperature. Experiments were performed using an Axopatch 1D (Molecular Devices) or a Dagan 3900A (Dagan Corp) patch clamp amplifier interfaced to a Dell microcomputer with a Digidata1332 analog/digital interface and the pCLAMP10 software package (Molecular Devices). For recording whole-cell $Na^+$ currents, pipettes contained (in mM): 5 NaCl, 90 $CsCH_3O_3S$, 20 CsCl, 1 $CaCl_2$, 10 EGTA, 10 HEPES, 4 MgATP, 0.4 Tris-GTP at pH 7.2, 300–310 mOsm. The bath solution contained (in mM): 20 NaCl, 110 TEACl, 10 CsCl, 1 $MgCl_2$, 1 $CaCl_2$, 10 HEPES, and 10 glucose, pH 7.4, 300–310 mOsm. See reference [60] for the experimental methods used to acquire the $I_{to,f}$ dataset and references [53,54] for those used to acquire the $I_{Na}$ HEK dataset.

### Evaluation of unbiased topologies to recapitulate canonical ion channel dynamics

A biophysically focused model topology was trained on $I_{to,f}$ human left ventricular myocytes, $I_{Na}$ voltage-clamp protocols in atrial mouse myocytes and $I_{Na}$ voltage-clamp protocols in HEK cells. Model rate constants were guaranteed to satisfy microscopic reversibility as outlined in Menon et al. (Eq 1). Values of *a* and *b* for each rate constant are listed in the S1 Appendix for $I_{to,f}$ and $I_{Na}$ example models. As mentioned in Menon et al.[16], the rate constants are exponential functions of steady state occupancies and rates (one-ion symmetrical barrier pore model) [61].

Parameters are optimized using an improved simulated annealing routine. The improved simulated annealing routine included multiple non-interacting chains to effectively "parallelize" and thereby speed the optimization process [62] and an adaptive temperature control scheme [43]. This temperature scheme begins at the lowest threshold temperature and is slowly incremented proportionally to the number of worse solutions encountered. When a "new" best solution is found, the temperature returns to lowest threshold. This scheme prevents the optimization from getting "stuck" in local optima:

$$t_i = (t_{min} + \lambda(\ln(1 + r_i))) \tag{2}$$

where $t_i$ is the current temperature is iteration $i$, $t_{min}$ is the minimum starting temperature, $\lambda$ is the temperature control parameter and $r_i$ is determined by the change in the best cost, $\Delta C$, at iteration $i$:

$$r_i = r_{i-1} + 1 \quad \Delta C > 0 \tag{3}$$

$$r_i = r_{i-1} \quad \Delta C = 0 \tag{4}$$

$$r_i = 0 \quad \Delta C < 0 \tag{5}$$

The cost function of the optimization was proportional to the sum of squared differences between each experiment and data value normalized by the experimental value plus a stiffness penalty:

$$cost = \sum \left( \frac{experiment - data}{data} \right)^2 + SP \tag{6}$$

A model stiffness penalty proportional to each optimized model's reciprocal condition number (1-norm) of the transition matrix, $A$, at varying voltages {1...N} was added to each model's cost:

$$rcond = \left( \|A(V_i)\|_1 \|A(V_i)^{-1}\|_1 \right)^{-1} \tag{7}$$

$$SP \sim \sum_{i=1}^{N} \quad rcond_i \; if \; rcond_i < threshold \tag{8}$$

This penalty preferentially selects models that do not require extremely small, computationally expensive, time steps when incorporated into cellular and tissue level excitability simulations. To lessen the risk of the optimal rate parameters depending on the initial starting conditions, the optimization included multiple starts (at least 20) with a quasi-random (Sobol) representation of the parameter space [44]. Quasi-random sequences increase coverage of the parameter space in each dimension (i.e. each free parameter is a dimension). Each sequential quasi-random sequence of dimension $D$ is mapped to the specified ranges for the $i^{th}$ parameter value:

$$parameter_i = (paramater_{i,max} - parameter_{i,min} * Sobol_{num[i]}) + parameter_{i,min} \tag{9}$$

Each start ran for at least 100,000 iterations beyond no change in 20% of cost unless terminated early for overfitting prevention. Because we expected time to convergence would depend on model complexity, the maximum optimization iterations allowed were periodically increased for convergence. To prevent overfitting in the simulated annealing optimizations, the cost function only applied to model values outside each of each data point's SEM and 20% of available experimental data was randomly set aside for validation. Quantitative metrics of progress and generalization loss were introduced to determine when to appropriately halt the optimization to prevent overfitting [46]. Progress quantifies how much the average training cost is larger than the minimum cost seen in the last k optimization iterations. Generalization loss quantifies how much larger the current validation cost is compared to the minimum validation cost seen previously. Three sequential increases in the ratio of the generalization loss to the progress results in early termination to prevent overfitting. We would certainly anticipate that more sophisticated validation sets will be available and will be used in future studies. Additional validation data will prove useful, along with an analysis of the best data to serve as

validation, in future iterations of this systematic method to minimize overfitting in models of higher complexity beyond those studied here.

## Training set of voltage-clamp protocols

**Steady state activation.** $I_{Na}$ HEK dataset: Steady-state probabilities were found at -100 mV. For voltages ranging between -45 mV to 20 mV, the peak open probability (a measure of peak current) was recorded after a step depolarization for 25 and normalized to maximum. $I_{to,f}$: Steady-state probabilities were found at -70 mV. For voltages between -60 and 60 mV in increments of 10 mV, the peak open probability (a measure of peak current) was recorded after a step depolarization for 50 ms and normalized to the maximum.

**Steady state inactivation.** $I_{Na}$ HEK dataset: Steady-state probabilities were found at -100 mV. A conditioning pulse at voltages between -110 mV to 40 mV in 10 mV increments was applied for 500 ms. The peak open probability (a measure of peak current) was then recorded after a test pulse at -10 mV for 25 ms and normalized to the maximum. $I_{to,f}$: Steady-state probabilities were found at -70 mV. Each preliminary voltage step in increments of 10 mV between -120 and 40 mV was held for 200 ms. The peak open probability (a measure of peak current) after a test pulse at 40 mV for 50 ms and normalized to the maximum.

**Recovery from inactivation.** $I_{Na}$ HEK: Steady-state probabilities were found at -100 mV. A depolarizing pulse at -10 mV for 500 ms was applied, followed by a hyperpolarizing pulse at -100 mV ranging between 0.5–210 ms. Peak current current was then recorded and normalized after a pulse at -10 mV for 25 ms. $I_{to,f}$: Steady-state probabilities were found at -70 mV. A depolarizing pulse at 40 mV for 500 ms was applied, followed by a hyperpolarizing pulse of -70 mV of variable time intervals (2–6000 ms). Peak current was then recorded and normalized after a pulse at 40 mV for 100 ms.

**Recovery from Use Dependent Block.** ($I_{Na}$ HEK only): Steady-state probabilities were found at -100 mV. A pulse train of a depolarization at -10 mV for 25 ms at 25 Hz was repeated for 100 pulses. A hyperpolarizing pulse at -100 mV for variable recovery intervals was applied for between 0.5–9000 ms. A test pulse followed at -10 mV for 25 ms and peak current was normalized to the maximum.

**Normalized current traces.** *(*$I_{to,f}$ only): Steady-state probabilities were found at -70 mV. Following a step depolarization to 20 and 60 mV for 10 ms, the normalized current was recorded at intervals of 0.2 ms.

**Deactivation Time Constants**. ($I_{Na}$ *only)*: Steady-state probabilities were found at -120 mV. Following recording the peak current after a depolarizing pulse at -20 mV for 5.0 ms, a hyperpolarizing voltage between -110 mV to -60 mV was applied for 5.0 ms and the time to 50% decay of peak current was recorded.

**Inactivation Time Constant**. ($I_{Na}$ HEK only): Steady-state probabilities were found at -100 mV. For voltages between -20 to 20 mV in 5 mV increments, the time to 50% decay of peak current was recorded.

**Maximum open probability.** $I_{Na}$ HEK and $I_{Na}$: To constrain open probabilities, maximum open probabilities of 0.27, 0.31, 0.29 at -20, -10, 0 mV, respectively (calculated from ten Tusscher 2006[52] solved in MATLAB with ode15s) were enforced. $I_{to,f}$: To best match the original $I_{to,f}$ simulated current, maximum open probabilities of 0.3 and 0.45 at 25 and 50 mV, respectively were enforced (calculated from [50], solved in MATLAB with ode15s)

## Validation set of voltage-clamp protocols

Twenty percent of experimental data was randomly chosen to serve as the validation set as is common when avoiding overfitting [63]. From each curve, 80% of the data points were

randomly selected and used to optimize the rate parameters. The remaining 20% of points were used to evaluate how well the model recapitulated the general trend in experimental data.

### Action potential model validation

Optimized models replaced respective currents in Tomek et al. [50] and ten Tusscher et al. [52] human ventricular action models. To simulate arrhythmogenic repetitive excitation for $I_{to,f}$ models, an action potential was elicited by an S1 stimulus followed by an S2 stimulus at various DI intervals following S1. To construct the dynamic restitution protocol, ten S1 beats at a BCL of 300 ms were simulated before the S2 stimulus at various DI intervals. Conduction velocity calculations for the $I_{Na}$ HEK models are from a 100-cell ten Tusscher [52] cable with reported velocities from Cell 50 after 10 beats at a BCL of 1000 ms.

### Computing resources

All simulation code was written in C++ and containerized using Docker to run on the Amazon Web Services Batch compute cluster. Model parsing code in Python and all code is available on GitHub [https://github.com/silvalab/AdvIonChannelMMOptimizer]. A sample $K^+$ conductance voltage optimization program is also available.

### Supporting information

**S1 Appendix. Supplementary Equations, Methods, and Example Model Rate Constants** (DOCX)

**S1 Fig. Illustrations of the maximum state degrees and long-range connections.** Green states indicate open states while blue indicates all other states **A)** A representative topology with a labeled state that has five possible connections to other states. By cutting out the red portion (containing the extraneous state and connection), the indicated state then has a degree of four**. B)** A representative model graph that has cycle length of six. Removal of the indicated red dashed connection breaks the cycle. The resulting topology has a maximum cycle length of four **C)** A representative 8-state topology that meets the maximum state and long-range connections restrictions illustrated in **A & B** and is included in the final count in **Fig 1E** (i.e. is an example of the 72,489 unique rooted topologies).
(TIF)

**S2 Fig. Illustration of progress and generalization loss measures stopping an optimization early as described in** [46]**. A)** Sample normalized training and validation costs during towards the end of an optimization. The training is slow, but steadily declining, while the validation cost is changing erratically at various optimization epochs. **B)** Measures of progress, generalization loss, and their ratio, Q, over the optimization time period as in **A**. Progress quantifies how much the average training cost is larger than the minimum cost seen in last k optimization iterations. Generalization loss quantifies how much larger the current validation cost is compared to the minimum validation cost seen across all iterations seen so far. In the example shown, progress (green dashes) stays relatively steady at various epochs reflecting the slow steady decline in training cost. Generalization loss widely fluctuates along with the validation error (red dashes). The ratio of the progress and generalization loss, Q (purple), steadily increases three simulation epochs in a row (as indicated by the black arrows), which signals that early stopping should occur.
(TIF)

**S3 Fig. Stiffness penalties for $I_{to,f}$ optimized models across all optimization starts. A)** The normalized stiffness penalty seen as across all starts as a function of increasing free rate constants. **B)** The number of steps in an explicit ODE solver (MATLAB's ODE45) when the penalties are not part of a model's cost. Only two models may be successfully solved with the less computationally intensive explicit solvers, which indicates the model solutions are inherently stiff. **C)** When including a measure of model stiffness in the optimization routine [48], more models can successfully be solved in the ODE45 routine.
(TIF)

**S4 Fig. Example $I_{to,f}$ model with 6 free rate constants A)** Topology of the model **B)** Experimental data points and simulated fits for conductance voltage and steady state inactivation protocols. Purple validation points indicate points added in the simulation to calculate a model's validation error when checking for overfitting. **C)** Experimental recovery from inactivation and simulated fit with added validation points to compute validation error for overfitting prevention. **D)** Normalized experimental current traces at 20 mV and 60 mV (black) with corresponding simulated traces. In purple is the experimental 40 mV normalized current trace along with its associated fit for validation. **E)** $I_{to,f}$ current trace when included in the Tor-Ord [50] human ventricular action potential model (**F**) All experimental data are from Johnson et al. [60] Experimental data points (black) are mean ± SEM.
(TIF)

**S5 Fig. State probabilities throughout recovery from inactivation protocol (for $I_{to,f}$) illustrates differences between acceptable and unacceptable models. A)** Recovery from inactivation protocol. **B)** State probabilities tracked as a function time throughout the above protocol. The 3-state model shown is representative of unacceptable $I_{to,f}$ models where recovery from inactivation is slow. Green designates the rooted open state while blue and red indicate the functional hypothetical inactivated state and closed state, respectively. At the resting state (-70 mV), the red "closed" state carries about 80% of the resting state probability while the blue "inactivated" state holds the other 20%. This closed state probability spread at the holding potential essentially "locks in" the voltage-dependent rates at -70 mV as slow. After the depolarizing pulse to +40 mV for 500 ms, the "blue" inactivated state carries 99% of the probability. The 25 ms hyperpolarizing pulse to -70 mV attempts to send the state probability back to the "red" closed state. However, the rate from the blue to red state at -70 mV is quite slow and, on reapplying the depolarizing step, the open state probability is lower than expected after 25 ms of recovery. **C)** An analogous representation of state probabilities for a 4-state model representative of the acceptable $I_{to,f}$ models where recovery from inactivation is appropriately fast. State colors are as before with the addition of 4th purple state. At the resting potential, the hypothetical closed state holds 99% of state probability. Therefore, the other states and rates are not "locked in" to slower rates at the holding potential. After the long depolarizing pulse, most state probability is again in the blue inactivated state. In this model, however, the rate from the inactivated state back to closed at -70 mV is fast enough to allow for sufficient recovery from inactivation after 25 ms hyperpolarization (77%). The pink state thus serves as a transitory state that distances the functionally inactive (blue) state from the open state (green).
(TIF)

**S6 Fig. Identification of possible structures for the $I_{Na}$ dataset. A)** Iterations completed for multiple optimizations starts of the models trained on the $I_{Na}$ dataset. Most optimization runs complete the maximum number of iterations allowed for the number of free rate constants, but as complexity increases, an optimization is more likely to be stopped early because of potential overfitting. **B)** Distribution of minimum normalized costs for each model with

multiple starts after completing optimization iterations as depicted in **A**. **C)** Representative acceptable models for $I_{Na}$. As defined in the text, acceptable models have minimum costs no larger than 300% of the absolute minimum cost. All models have at least 8 free rate constants. **D)** Representative unacceptable models. All models have minimal states and/or sparsely connected with a range of free rate constants. **E)** -**G)** Representative voltage-clamp models fits for steady state activation, inactivation, recovery from inactivation and current traces for models in the acceptable and unacceptable model categories. **H)** Representative ten Tusscher human ventricular action potentials [52] with replaced modeled acceptable and unacceptable $I_{Na}$ currents.
(TIF)

**S7 Fig. Stiffness penalties for $I_{Na}$ optimized models across all optimization starts. A)** The normalized stiffness penalty seen as across all starts as a function of increasing free rate constants. There is decline in the minimum stiffness penalty from four to six free rate constants followed by a plateau in the minimum penalty for topologies with greater than six free rate constants **B)** The number of steps in an explicit ODE solver (MATLAB's ODE45) when the penalties are not part of model cost. Few models may be successfully solved with the less computationally intensive explicit solvers, which indicates the model solutions are inherently stiff. **C)** When including a measure of model stiffness in the optimization routine, more models can successfully be solved in the ODE45 routine.
(TIF)

**S8 Fig. Example model for $I_{Na}$ with 9 free rate constants. A)** Topology of the model and **B)** Experimental data points and simulated fits for conductance voltage and steady state inactivation protocols. Purple validation points indicate points added in the simulation to calculate a model's validation error when checking for overfitting. **C)** Experimental recovery from inactivation and simulated fit with added validation points to compute validation error for overfitting prevention. **D)** Normalized experimental current amplitudes recorded at -10 mV and 10 mV, plotted as points (black) with corresponding simulated traces in green. The normalized experimental current data acquired at 0 mV along with its associated fit for validation is shown in purple. **E)** Modeled $I_{Na}$ current when inserted into the ten Tusscher [52] human ventricular action potential model and the resulting action potential **F)**. Experimental data points (in black) in **B)**, **C)** and **D)** are means ± SEM.
(TIF)

**S9 Fig. Summary of Markov model performance for all $I_{Na}$ topologies studied (as in Fig 4C and 4D) with unacceptable model solutions included.** Unacceptable and acceptable labels are as previously defined. Tentative yellow topologies produced acceptable voltage protocol fits but did not perform like other acceptable models in the action potential validation (different degrees of repolarization failure).
(TIF)

**S10 Fig. Dynamic restitution curve variability of all acceptable $I_{to,f}$ models. A)** Boxes Labeled B and C correspond to the regions of the restitution curve of shallower **B)** and steeper **C)**, respectively. **B)** Left: Acceptable model voltage protocol fits. Right: Voltage and $I_{to,f}$ traces following the 10th S1 stimulus followed by a S2 stimulus at various DI intervals. $I_{to,f}$ inactivates completely in 100 ms, resulting in a shallower restitution slope between 60–80 ms DI intervals. **C)** Left: Another acceptable model voltage protocol fits. Right: Voltage and $I_{to,f}$ traces during the 10th S1 stimulus followed by a S2 stimulus at various DI intervals. In this acceptable model, $I_{to,f}$ does not inactivate until 200 ms, resulting in a steeper restitution slope between DI

intervals 60–80 ms.
(TIF)

**S11 Fig. Variability in activation timing of acceptable I<sub>Na</sub> HEK models.** Both representative models show acceptable I$_{Na}$ HEK voltage-clamp fits but the activation timing differs: 4 ms **(A)** versus 8 ms **(B)** **(See Fig 5B).**
(TIF)

**S12 Fig. Activation and Inactivation Variability in Acceptable I$_{Na}$ HEK models. A)** Time to peak at two voltages -45 mV and 0 mV starting from a holding potential of -86.2 mV as in ten Tusscher [52]. Inset: Zoomed in version of A between 0–0.5 ms. **B)** Conduction velocity vs inactivation of acceptable I$_{Na}$ models with time to peak between 0.07–1.00 ms at both -45 mV and 0 mV to control for the extent of activation. Inactivation is the maximum open probability ratio at -20 mV for 10 ms following two conditions: 1) -90 mV at steady state, 1000 ms at 40 mV, -90 mV for 10 ms 2) hold at -90 mV.
(TIF)

## Acknowledgments

The authors would like to acknowledge Michael Southworth for programming guidance.

## Author Contributions

**Conceptualization:** Kathryn E. Mangold, Jonathan R. Silva.

**Data curation:** Kathryn E. Mangold, Wei Wang, Eric K. Johnson.

**Formal analysis:** Kathryn E. Mangold, Druv Bhagavan, Jonathan R. Silva.

**Funding acquisition:** Jeanne M. Nerbonne, Jonathan R. Silva.

**Methodology:** Kathryn E. Mangold, Jonathan R. Silva.

**Project administration:** Kathryn E. Mangold, Jonathan R. Silva.

**Resources:** Kathryn E. Mangold, Jeanne M. Nerbonne, Jonathan R. Silva.

**Software:** Kathryn E. Mangold.

**Supervision:** Kathryn E. Mangold, Jeanne M. Nerbonne, Jonathan R. Silva.

**Validation:** Kathryn E. Mangold, Druv Bhagavan, Jonathan D. Moreno.

**Visualization:** Kathryn E. Mangold, Druv Bhagavan.

**Writing – original draft:** Kathryn E. Mangold, Jonathan R. Silva.

**Writing – review & editing:** Kathryn E. Mangold, Jonathan D. Moreno, Jeanne M. Nerbonne, Jonathan R. Silva.

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
