## [Decision Letter · Decision Letter 0]

30 Apr 2021

Dear Dr. Silva,

Thank you very much for submitting your manuscript "Identification of Structures for Ion Channel Kinetic Models" for consideration at PLOS Computational Biology.

As with all papers reviewed by the journal, your manuscript was reviewed by members of the editorial board and by several independent reviewers. In light of the reviews (below this email), we would like to invite the resubmission of a significantly-revised version that takes into account the reviewers' comments.

We cannot make any decision about publication until we have seen the revised manuscript and your response to the reviewers' comments. Your revised manuscript is also likely to be sent to reviewers for further evaluation.

Sincerely,

Jeffrey J. Saucerman

Associate Editor

PLOS Computational Biology

Daniel Beard

Deputy Editor

PLOS Computational Biology

Reviewer's Responses to Questions

**Comments to the Authors:**

Reviewer #1: In this study by Mangold and colleagues, the authors demonstrate a generalized approach to identify acceptable Markov chain models representing ion channels kinetics. As the authors note, this is a critical challenge in the electrophysiology field, as most modeling approaches consider fits of rate constant parameters but provide much less consideration of the Markov chain model “structure” or “topology.” The authors systematically identify model topologies with reasonable biophysical constraints and determine acceptable and unacceptable fits to ion channel data for two different ion channels.

Overall, I find the study to be rigorously performed and the manuscript to be well written. I have a few specific comments for the authors to address.

Major:

1. The most significant issue that I see is that many of the general conclusions given regarding Markov chain topology and acceptable/unacceptable fits are described qualitatively. However, the authors have the data to provide quantitative measurements. (There are some specific numbers in the Discussion provided but not more generally for all cases.)

Specifically, as an example, the dots in Figure 4 provide a general sense of how the number of states and edges and root degree impact acceptable, tentative, and unacceptable fits. Please provide some quantitative measurements of this data, e.g., how does the percentage of acceptable fits change as the network properties changes?

2. The authors illustrate nicely the differences in acceptable vs unacceptable fits, in terms of fitting the data and also matching action potential characteristics. However, it is less clear how different or similar multiple “acceptable” fits are, specifically when incorporated into cellular and/or tissue models and tested outside the conditions in which the fits are performed.

For example, for the Ito,f current, the authors illustrate the fit to a specific S1S2 pacing protocol with a 200 ms S2 interval. If the authors measured a dynamic APD restitution curve with different Ito,f models in a ventricular cell model, how much variation would there be between different “acceptable” fits? Do the differences depend on whether or not the model is a more complex or simpler model?

For the INa current, for example, are there differences in conduction velocity (CV) and/or CV restitution for different “acceptable” fits in a 1D tissue, and if so, is there an explanation for these differences?

More generally, are “acceptable” fits for the same network topology more similar, compared with fits from different topology?

The authors address some of these issues in the Discussion, noting that different “acceptable” fits are practical for different situations, and I appreciate that answering the above questions for all “acceptable” fits may not be a computationally feasible exercise, in particular for the tissue simulations, but providing some examples would greatly strengthen the paper.

Minor:

1. There is a grammatical issue with the sentence that starts at line 65. The authors likely mean “Both cases…” instead of “In both cases”

2. Lines 391-2: The authors state there 167 distinct topologies but only 169 acceptable model solutions. Is one of these numbers a typo?

3. In several of the figures, a red asterisk is placed next to some of the network topologies. What does this represent? Please include this detail in the figure caption.

4. Some aspects of the optimization procedure are difficult to follow, as described in the Results and Methods. In particular, I did not understand the explanation of progress, generalization loss, and Q as shown in Figure S2.

Reviewer #2: In the paper by Mangold and co-authors, the authors set out to develop a systematic approach to identify topologies for Markov model representations of ion channels. This is a great paper. The authors demonstrate a potential solution to a long-standing problem: how to identify the appropriate Markov state model that best represents ion channel gating for a particular ion channel species.

One aspect of this paper that is particularly innovative, is that the authors utilize biophysical properties of ion channels to restrict the number of available topologies. This is an advantage over standard combinatorial methods, which generally results in large numbers of models that have no physiological basis in reality.

I really didn’t find anything not to like in this paper. I think the methodology is useful, will be widely adopted, and it’s a starting point that will allow for continued improvement. Figures are clear and the writing is excellent and manages to take a complex topic and make it accessible.

My major comment is that I hope the source code utilized in the paper can be made freely available. I didn’t see a link to the source code on the github, but would ask that the authors make that available to allow the whole process to be utilized by others. User friendly will be the key here - please make sure the codes are well documented and user friendly.

Reviewer #3: This paper attempts an exhaustive enumeration and validation of multi-state kinetic models of voltage gated ion channels, evaluated for quality of fit to data for several specific channels from cardiac myocytes.

I am very sympathetic to the aims of this study. I think this is a timely and important problem. Robust and principled approaches to identifying ion channel models are needed, and in my view this study goes some way towards achieving this goal.

My main critique of the paper is in its presentation. With apologies to the authors for my bluntness, large parts of the paper read as a project progress report (... first we did this, next we did that....). The language is often quite informal and imprecise. There is frequent use of vague qualifiers, such as 'a reasonable number of ...', '...still quite large.' Such vague statements are not appropriate. Furthermore, there are statements made for which no justification is given. As an example, page 5, 'These rates would be increasingly difficult to identify experimentally as the number of connections increases.' No justification for this statement is seemingly provided in the text. This is one of many such instances. I suggest the authors reread their text and where such statements are made, provide justification or remove. Finally, large sections of the manuscript have no subheadings to help guide the reader, and were hard to navigate and in all honesty rather monotonous as a consequence: in particular the Results section and the Discussion and Future Directions section. I would strongly encourage the authors to clarify what are the major points that they wish to get across, and shorten and subdivide / subtitle these sections accordingly.

There is also a problem with the current order of the paper, in that some concepts that are in my view critical to understanding what the authors are doing are presented towards the end in the method section. In particular, the actual model being used (the form of the rate constants including voltage dependence, for example) doesn't get discussed until then. In order for the presentation to be intelligible these concepts (but not specific implementation details) need to come earlier (perhaps even in the introduction, given that they were taken from earlier work).

Some questions that may be of interest to address: I have never understood why ion channel models only consider one open state. Is there a good reason for this? If so it would be helpful to mention it.

Cycles are highly problematic in this type of model unless adequately dealt with, for two reasons. Firstly equilibrium constraints mean that for each and every cycle there is a parameter dependence that has to be satisfied (I think the authors mention this, again not until the methods - this needs to be mentioned earlier). Previous authors have also found that even when these constraints are made explicit, parameter identification with cycles is problematic (presumably because of difficulties with unique parameter identification). Did the authors observe any differences between models with / without cycles when fitting to data?

The authors do not give much attention to the question as to the data needed to identify models. Effectively what the approach outlined in this manuscript will ideally determine is what is the best model given the data. This of course raises the issue as to whether other models might actually be better if more data or different data were available. I wonder whether the authors have any thoughts on this other dimension to the model identification challenge here?

I was confused by Figure 1A. It isn't clear to me whether this is trying to illustrate the unique topologies with three nodes, or trying to illustrate different ways in which topologies may be considered repeats in respect of ion channel models. On the top line the first and second models are clearly different, but the third is a relabled version of the first. How is this indicated through the highlighting was not clear to me, especially as further down in the first line with cyclic models, the two pink highlighted models now do appear to be the same.

Finally, are there any insights that can be gained into the biology of ion channels that arise from trying to relate the topologies that are seemingly preferred in this methodology to known biophysical characteristics of ion channel proteins? In particular this question is raised because, perhaps naively, I would have expected models with topologies similar to those shown in Figure 3B to be more the commonly accepted topologies (including as determined in for single channel data), and would have thought models with more cycles as in Figure 3A, to be less representative of current modelling approaches.

**Have the authors made all data and (if applicable) computational code underlying the findings in their manuscript fully available?**

Reviewer #1: Yes

Reviewer #2: **No: **please make code for all process available.

Reviewer #3: Yes

PLOS authors have the option to publish the peer review history of their article (what does this mean?). If published, this will include your full peer review and any attached files.

Reviewer #1: **Yes: **Seth H. Weinberg

Reviewer #2: **Yes: **Colleen Clancy

Reviewer #3: No
---

## [Decision Letter · Decision Letter 1]

16 Jul 2021

Dear Dr. Silva,

We are pleased to inform you that your manuscript 'Identification of Structures for Ion Channel Kinetic Models' has been provisionally accepted for publication in PLOS Computational Biology.

Best regards,

Jeffrey J. Saucerman

Associate Editor

PLOS Computational Biology

Daniel Beard

Deputy Editor

PLOS Computational Biology

Reviewer's Responses to Questions

**Comments to the Authors:**

Reviewer #1: The authors have nicely addressed all of my concerns. I have no further concerns.

Minor: In the new Figure 5B, the blue Na+ current trace aligns in time with the orange voltage trace, and the orange Na+ current traces aligns in time with the blue voltage trace, so I suspect that the colors of the voltage traces have been switched.

Reviewer #2: Superb study all around - authors have done well in responding to all earlier concerns raised by all reviewers.

**Have the authors made all data and (if applicable) computational code underlying the findings in their manuscript fully available?**

Reviewer #1: Yes

Reviewer #2: Yes

PLOS authors have the option to publish the peer review history of their article (what does this mean?). If published, this will include your full peer review and any attached files.

Reviewer #1: **Yes: **Seth H. Weinberg

Reviewer #2: **Yes: **Colleen E Clancy

---

## [Editor Report · Acceptance letter]

6 Aug 2021

PCOMPBIOL-D-21-00592R1 

Identification of Structures for Ion Channel Kinetic Models

Dear Dr Silva,

I am pleased to inform you that your manuscript has been formally accepted for publication in PLOS Computational Biology. Your manuscript is now with our production department and you will be notified of the publication date in due course.

With kind regards,

Andrea Szabo
